# Dueling Convex Optimization with General Preferences

**Aadirupa Saha** [1]   **Tomer Koren** [2][3]   **Yishay Mansour** [2][3]

## Abstract

We address the problem of convex optimization with dueling feedback, where the goal is to minimize a convex function given a weaker form of *dueling* feedback. Each query consists of two points and the dueling feedback returns a (noisy) single-bit binary comparison of the function values of the two queried points. The translation of the function values to the single comparison bit is through a *transfer function*. This problem has been addressed previously for some restricted classes of transfer functions, but here we consider a very general transfer function class which includes all functions that admit a series expansion about the origin. Our main contribution is an efficient algorithm with convergence rate of $O(\epsilon^{-4p})$ for smooth convex functions, and an optimal rate of $\widetilde{O}(\epsilon^{-2p})$ when the objective is both smooth and strongly convex, where $p$ is the minimal degree (with a non-zero coefficient) in the transfer's series expansion about the origin.

## 1. Introduction

Convex optimization algorithms are fundamental across many fields, including machine learning. Most commonly, convex optimization is studied in a first-order gradient oracle model, where the optimization algorithm may query gradients of the objective function; or a more limited model of zero-order oracle access, where the optimization algorithm may only query function values. Such optimization frameworks are well-studied in the literature (e.g., Nesterov, 2003). However, in recent days, relative/preference-based query models gained much attention, which are common in domains such as recommendation systems, online merchandises, search engine optimization, crowd-sourcing, drug testing, tournament ranking, social surveys, etc. (Hajek et al.,

2014; Khetan and Oh, 2016).

Drawing motivation from the above, in this paper we study a challenging convex optimization model where the access to the objective function is through a *noisy pairwise comparison oracle*. Namely, given an underlying convex objective function $f : \mathbb{R}^d \mapsto \mathbb{R}$, at each step the optimization algorithm is allowed to query two points $\mathbf{w}, \mathbf{w}'$ in $\mathbb{R}^d$, upon which only a noisy 1-bit feedback $o_t \in \{\pm 1\}$ is revealed, whose expected value indicates their relative function values. More specifically, the feedback signal $o_t$ is such that

$$\mathbb{E}[o_t \mid \mathbf{w}, \mathbf{w}'] = \rho(f(\mathbf{w}) - f(\mathbf{w}')),$$

where $\rho : \mathbb{R} \mapsto [-1, 1]$ is a (possibly nonlinear) *transfer function* mapping difference in function values to a signed preference signal, and $\rho(f(\mathbf{w}) - f(\mathbf{w}'))$ is interpreted as the degree to which $\mathbf{w}$ should be preferred over $\mathbf{w}'$, or vice versa. Provided such access, our goal is to find a point that approximately minimizes the objective $f$. Borrowing terminology from the literature on dueling bandits, we call our framework *General Dueling Convex Optimization* (G-DCO) for general transfer functions.

Noisy pairwise comparison access could potentially be significantly weaker than the already weak zero-order access. Indeed, a special case of this framework has been studied by Jamieson et al. (2012) who focused on polynomial transfer functions of the form $\rho(x) = c\,\text{sign}(x)|x|^p$ and gave tight upper and lower bounds in the pairwise comparison model for *strongly convex and smooth* objectives. Their results indicate that as $p$ grows larger, the best achievable convergence rate degrades quickly, and already when $p > 1$ this rate becomes strictly inferior to that of zero-order optimization. Much more recently, Saha et al. (2021b) considered a similar pairwise comparison model with a different type of a transfer function, namely the sign function $\rho(x) = \text{sign}(x)$ (i.e., $p = 0$), and established fast convergence rates for this case exclusively.

These works point us to some fundamental questions: Can we design algorithms for dueling convex optimization that is able to leverage more general transfer functions? Can we converge to a minimizer even when the transfer is unknown to the algorithm? And what properties of the transfer function dictate the achievable optimization rates? We make progress toward answering these questions:

[1]Department of Computer Science, University of Illinois, Chicago, US [2]Blavatnik School of Computer Science, Tel Aviv University, Israel [3]Google Research, Tel Aviv, Israel. Correspondence to: Aadirupa Saha <aadirupa@uic.edu>.

*Proceedings of the 42$^{nd}$ International Conference on Machine Learning*, Vancouver, Canada. PMLR 267, 2025. Copyright 2025 by the author(s).

(i) We formalize a generalized dueling convex optimization setting for convex optimization with pairwise-preference feedback given by a general transfer function $\rho$, which is only assumed to be well-behaved around the origin (see Section 2.1 for a precise definition of the query model and optimization objective). Our framework generalizes and significantly extends two existing settings of optimization with comparison feedback of Jamieson et al. (2012) and Saha et al. (2021b).

(ii) We give a novel algorithm for dueling convex optimization with a general transfer function $\rho$, called Projected Dueling Descent (Algorithm 1). We prove that when the optimization objective function is convex and smooth, our algorithm needs an order of $O(\epsilon^{-4p})$ queries to the pairwise-preference oracle for finding an $\epsilon$-optimal point (see Theorem 2); here, $p$ can be thought of the minimal non-zero degree in a series expansion of the transfer $\rho$ around zero.

(iii) We further show that our algorithm can achieve faster convergence rates when the function is additionally also strongly convex (Algorithm 2, Theorem 4). Concretely, we show that in this case only $\widetilde{O}(\epsilon^{-2p})$ pairwise queries are sufficient for $\epsilon$-convergence. The latter rate is shown to be tight as it matches existing lower bounds (for certain transfer functions) for strongly convex optimization with comparison feedback due to Jamieson et al. (2015). We remark that this result does not follow via a reduction to the general convex case, as adding (strongly convex) regularization to the objective breaks the comparison oracle access.[1]

We emphasize that, in fact, even the most well-studied transfers, like the sigmoid, were not covered by the existing literature. The goal of this paper is to fill in this gap, significantly expanding the scope of comparison-based (dueling) optimization to a variety of nonlinear transfer functions.

Our algorithmic results complement those of Saha et al. (2021b), who only considered the sign transfer function. Compared to the results of Jamieson et al. (2012), we are able to handle both the convex and strongly convex cases (while they only deal with the latter), and we only require the transfer to be well-behaved around the origin (while they rely on its global structure[2]). Thus, we are able to encompass a greater variety of transfer functions whose local behavior around zero is approximated by a polynomial—this

includes virtually all functions that admit a series expansion around the origin.

In terms of techniques, our approach starts by following standard lines in bandit and zero-order optimization (e.g., Flaxman et al., 2005; Saha and Tewari, 2011), forming stochastic estimates for the gradients of the underlying objective using the available dueling access. However, unlike in the standard convex scenarios, the gradient estimates we are able to construct in the dueling setting are substantially deteriorated by the nonlinearity of the transfer: information about the magnitude of the gradient is badly mutated by the feedback process, and even further, the gradient estimates closer to optimality are biased and thus hamper convergence. Our analysis addresses the first issue by carefully analyzing how the gradient estimates are skewed by the nonlinear transfer function, through inspecting its behavior around the origin and how it transforms the expected value of our estimates. The second issue turns out to be even more challenging to address technically, since the stochastic process induced by the optimization trajectory incurs a "conditional drift" close to where we would like to establish its convergence. We circumvent this challenge through a stopping-time analysis of a supermartingale related to this stochastic process, that establishes its concentration with high probability.

## 2. Preliminaries

**Notation.** Let $[n] = \{1, 2, \ldots n\}$, for any $n \in \mathbb{N}$. Given a set $S$, for any two items $x, y \in S$, we denote by $x \succ y$ the event $x$ is preferred over $y$. For any $r > 0$, let $\mathcal{B}_d(r)$ and $\mathcal{S}_d(r)$ denote the ball and the surface of the sphere of radius $r$ in $d$ dimensions, respectively. $\mathbf{I}_d$ denotes the $d \times d$ identity matrix. For any vector $\mathbf{x} \in \mathbb{R}^d$, $\|\mathbf{x}\|_2$ denotes the $\ell_2$ norm of vector $\mathbf{x}$. We write $\tilde{O}$ for the big O notation up to logarithmic factors.

### 2.1. Problem Setup

We consider the problem of minimizing a convex function $f : \mathbb{R}^d \mapsto \mathbb{R}$ over a bounded convex domain $\mathcal{D} \subseteq \mathbb{R}^d$. We assume that $f$ is $G$-Lipschitz and $\beta$-smooth over $\mathcal{D}$, and that $\mathcal{D}$ has Euclidean diameter bounded by $D$. We denote by $\mathbf{w}^* \in \arg\min_{\mathbf{w} \in \mathcal{D}} f(\mathbf{w})$ a point where $f$ is minimized over $\mathcal{D}$.

***Query model:*** Our access to the objective $f$ is through a noisy comparison oracle that upon input of a pair $(\mathbf{w}, \mathbf{w}')$ of points in $\mathbb{R}^d$, emits a random binary response $o \in \{\pm 1\}$ such that

$$\mathbb{E}[o \mid \mathbf{w}, \mathbf{w}'] = \rho(f(\mathbf{w}) - f(\mathbf{w}')),$$

where $\rho : \mathbb{R} \to [-1, 1]$ is a fixed *transfer function* mapping difference in function values to (signed) preferences, unknown to the algorithm.

---

[1] In other words, in general one cannot implement a comparison oracle for $f(x) + \frac{\alpha}{2}\|x\|^2$ given a black-box comparison oracle for $f$.

[2] Indeed, their algorithm relies on a line-search procedure at each step, employing the comparison oracle for implementing a one-dimensional noisy binary search.

For example, given $\rho$ the query model could output a random variable $o$ such that $o \sim \mathrm{Ber}^{\pm}\big(\rho(f(\mathbf{w}) - f(\mathbf{w}'))\big)$ where $\mathrm{Ber}^{\pm}$ denotes a signed version of the Bernoulli distribution (such that for a random variable $X \sim \mathrm{Ber}^{\pm}(p)$, we have $\Pr(X = +1) = 1 - \Pr(X = -1) = \frac{p+1}{2}$).

***Transfer function:*** We assume throughout that the transfer function $\rho : \mathbb{R} \mapsto [-1, 1]$ is fixed and satisfies the following assumptions:

**Assumption 1.** *(i) The transfer $\rho$ is differentiable, monotonically non-decreasing and satisfies $\rho(0) = 0$; (ii) there are constants $r > 0, c_\rho > 0$ and integer $p \geq 1$ such that for all $x \in (-r, r)$ it holds that $\rho'(x) \geq c_\rho p|x|^{p-1}$.*

We discuss these assumptions in more detail in the subsection below, where we argue that they hold for essentially any function $\rho$ that admits a series expansion about the origin.

***Optimization goal:*** The goal of the optimization process is then, given $\epsilon > 0$, to find a point $\mathbf{w} \in \mathcal{D}$ such that $f(\mathbf{w}) - f(\mathbf{w}^*) \leq \epsilon$ while minimizing the number of queries to the comparison oracle.

To formalize our optimization framework further, we now introduce the notion of admissible transfer functions, a broad class of comparison mechanisms characterized by smoothness and monotonicity, which includes most functions expressible via a series expansion around the origin.

## 2.2. Admissible Transfer Functions

The second part of Assumption 1 is perhaps the most stringent one; however, in the following lemma we demonstrate that it is satisfied by a wide variety of natural transfer functions: those that admit a series expansion about the origin with uniformly bounded coefficients.

**Lemma 1.** *Let $\rho$ admit a series expansion $\rho(x) = \sum_{n=p}^{\infty} a_n x^n$ about $x = 0$ with minimal degree $p \geq 1$ and radius of convergence $\delta > 0$. Then, if $a_p > 0$ and $|na_n| \leq M$ for all $n > p$, we have that*

$$|\rho'(x)| \geq \tfrac{1}{2} p a_p |x|^{p-1} \quad for \quad |x| < \min\Big\{\delta, \frac{pa_p}{4M}\Big\}.$$

Note that since we require $\rho(0) = 0$, it must be that $a_0 = 0$ and the assumption $p \geq 1$ holds naturally. Further, since we would like $\rho(x) > 0$ to hold for $x > 0$, the first nonzero coefficient must be positive, namely $a_p > 0$. Thus, the only non-trivial assumption is that the series coefficients are uniformly bounded; however, this condition holds for many natural transfer functions: e.g., for the sigmoidal $\arctan(x)$, hyperbolic tangent $\tanh(x)$ and for the error function $\mathrm{erf}(x)$, it holds simply with $M = 1$.

*Proof of Lemma 1.* On the interval of convergence $(-\delta, \delta)$ we have $\rho'(x) = \sum_{n=p}^{\infty} na_n x^{n-1}$ as one can exchange

the order of summation and differentiation. Let us write $\rho'(x) = pa_p x^{p-1} + R(x)$, where $R(x) = \sum_{n>p} na_n x^{n-1}$. Then, for $|x| < \delta \leq \frac{1}{2}$, we can show that: $|R(x)| \leq$

$$\sum_{n>p} |na_n| \, |x|^{n-1} \leq M|x|^p \sum_{n=0}^{\infty} |x|^n$$
$$= M|x|^p \frac{1}{1 - |x|}$$
$$\leq 2M|x|^p.$$

Thus, when $|x| \leq pa_p/4M$ we have $|R(x)| \leq \frac{1}{2}pa_p|x|^{p-1}$. It follows that $|\rho'(x)| \geq pa_p|x|^{p-1} - |R(x)| \geq \frac{1}{2}pa_p|x|^{p-1}$ as claimed. $\qquad\square$

## 3. Dueling Convex Optimization with General Transfer Functions

In this section we propose an optimization algorithm for dueling optimization problem for a convex and $\beta$-smooth objective $f$. The primary difficulty in designing an efficient algorithm for the purpose lies in the fact that we cannot hope to estimate the gradient of $f$ for any general dueling/pairwise preference model (i.e., any general $\rho$). Thus, we cannot directly apply standard gradient estimation techniques (e.g., Flaxman et al., 2005; Saha and Tewari, 2011) to address this problem.

We get around the difficulty by noting that, though one may not be able to estimate the exact gradient of $f$, $\nabla f(\mathbf{w})$, at a given point of interest $\mathbf{w} \in \mathcal{D}$, we can hope to estimate a '$p$-th order proxy of $\nabla f(\mathbf{w})$', called "relative gradient" of $f$ at $\mathbf{w}$, from the 1-bit preference feedback generated according to the transfer function (or pairwise preference model) $\rho$. The following definition and the lemma describe a more formal argument for this.

### 3.1. The Projected Dueling Descent algorithm

The crux of the idea lies in designing "relative gradient" based algorithm (Algorithm 1), which is a generalized notion of gradient descent-based optimization technique: The algorithm proceeds sequentially, where at each step $t$, it maintains a solution $\mathbf{w}_t \in \mathcal{D}$, estimates the "relative gradient" of $f$ at point $\mathbf{w}_t$ using dueling feedback, and take a small step in the negative direction of the estimated "relative gradient" to reach the updated solution $\mathbf{w}_{t+1}$.

More formally, the algorithm starts from an initial point $\mathbf{w}_1 \in \mathcal{D}$. At any round $t = 1, 2, \ldots$, the algorithm queries the dueling feedback on a pair of points $(\mathbf{w}_t + \gamma \mathbf{u}_t, \mathbf{w}_t - \gamma \mathbf{u}_t)$, where $\mathbf{u}_t \sim \mathrm{Unif}(\mathcal{S}_d(1))$ is a random unit vector, and $\gamma$ is a perturbation scale parameter.[3] Upon receiving the 1-bit preference feedback $o_t \in \{\pm 1\}$, it finds a "relative

---

[3]Recall that our setup allows for the oracle to process queries of

gradient" an estimate of $f$ at $\mathbf{w}_t$ as $\mathbf{g}_t := o_t \mathbf{u}_t$ which gives a valid descent direction on expectation. It then takes a step, with stepsize $\eta$, along the negative direction of $\mathbf{g}_t$ to obtain the next iterate $\mathbf{w}_{t+1} := \mathbf{w}_t - \eta \mathbf{g}_t$ (with a suitable projection if necessary).

The details of the algorithm are presented in Algorithm 1. Note that the algorithm itself is oblivious to the specific transfer function $\rho$ and does not require any knowledge about it or its parameters.[4]

---

**Algorithm 1 Projected Dueling Descent (PDD)**

1: **Input:** Initial point: $\mathbf{w}_1 \in \mathcal{D}$, Learning rate $\eta$, Perturbation parameter $\gamma$, Query budget $T$
2: **for** $t = 1, 2, 3, \ldots, T$ **do**
3:     Sample $\mathbf{u}_t \sim \mathrm{Unif}(\mathcal{S}_d(1))$
4:     Set $\mathbf{x}'_t := \mathbf{w}_t + \gamma \mathbf{u}_t$, $\mathbf{y}'_t := \mathbf{w}_t - \gamma \mathbf{u}_t$
5:     Play the duel $(\mathbf{x}'_t, \mathbf{y}'_t)$, and observe $o_t \in \pm 1$ such that $o_t \sim \mathrm{Ber}^{\pm}\big(\rho\big(f(\mathbf{x}'_t) - f(\mathbf{y}'_t)\big)\big)$.
6:     Update: $\tilde{\mathbf{w}}_{t+1} = \mathbf{w}_t - \eta \mathbf{g}_t$, where $\mathbf{g}_t = o_t \mathbf{u}_t$
7:     Project: $\mathbf{w}_{t+1} = \arg\min_{\mathbf{w} \in \mathcal{D}} \|\mathbf{w} - \tilde{\mathbf{w}}_{t+1}\|$
8: **end for**

---

On a high level, following prior work (Saha et al., 2021b), Algorithm 1 takes the approach of designing a gradient estimator using the dueling feedback and using it for performing gradient descent. However, our gradient estimation analysis below is very different due to the generality and nonlinearity of the transfer function in the feedback. This will be seen in Lemma 3 and its proof, which analyzes the gradient estimator in the case of nonlinear transfer: roughly, we inspect the local polynomial behavior of the transfer around zero and work out the effect of the polynomial transformation on the expected value of the gradient estimator. In the estimation process, the magnitude of the gradient gets distorted (see the $\|\nabla f(w_t)\|^{p-1}$ leading factor in the expected value) so we can no longer use the standard subgradient descent analysis, which we replace with a different approach akin to the analysis of normalized gradient descent.

### 3.2. Convergence analysis

We now state the main result of this section, which provides a convergence guarantee for our algorithm for convex and smooth objectives.

**Theorem 2.** *Assume that the objective $f$ is convex, $\beta$-*

---

[4]points that lie outside of the domain $\mathcal{D}$ (we assume $f$ is defined on the entire space, yet optimization is constrained to $\mathcal{D}$). Technically, we can limit queries to a slightly dilated version of the domain $\mathcal{D}$; we avoid these nuicanses here.

[4]That said, as our theorem below shows, the optimal tuning of the learning rate $\eta$ does depend on the underlying problem parameters, but this is the case with essentially any optimization algorithm with a learning rate parameter, including gradient descent, stochastic gradient descent, etc.

*smooth and $G$-Lipschitz over $\mathcal{D}$. For any $0 < \epsilon < \bar{\epsilon} = 5\beta \min\{dD^2, \sqrt{d}rD/G\}$ and $\delta \in (0,1)$, running Algorithm 1 with*

$$\gamma = \frac{\epsilon}{10\beta D\sqrt{d}}, \quad 0 < \eta \le \frac{c_\rho p \epsilon^{2p}}{\beta^p (80D)^{2p-1} d^{p+1/2}},$$

*for $T > 2(D^2/\eta^2 + 1)\log(1/\delta)$ steps, certifies that with probability at least $1-\delta$, there is at least one step $1 \le t \le T$ where $f(\mathbf{w}_t) \le f(\mathbf{w}^*) + \epsilon$.*

Note that the theorem's statement applies to sufficiently small error $\epsilon < \bar{\epsilon}$; for larger values the theorem can be used with $\epsilon = \bar{\epsilon}$, in which case the convergence rate only depends on the problem parameters (and is independent of $\epsilon$).

Theorem 2 gives a sample complexity bound of $O(\epsilon^{-4p})$ for reaching $\epsilon$-optimality, with high probability; put differently, it establishes an $O(T^{-1/4p})$ convergence rate over $T$ steps. A notable limitation of the convergence guarantee, however, is that it only implies that one of the points we evaluate along the trajectory of the algorithm is $\epsilon$-optimal with high probability, rather than specifying a particular point as output. (We note that this issue appears only in the non-strongly-convex case: indeed, as we show in the next section, in the strongly convex case our algorithm produces a single $\epsilon$-optimal point with high probability.)

We can address this issue and detect an $\epsilon$-optimal point using additional comparison queries to the noisy comparison oracle. Specifically, given $\mathbf{w}_1, \ldots, \mathbf{w}_T$ we can construct a complete binary tree whose leaves are the $T$ points; each internal node in the tree evaluates the (noisy) min of its descendants, requiring roughly $\epsilon^{-2p}$ queries to the oracle. Since errors accumulate only along the path from the root to the true minimum, both the accuracy and confidence of the entire comparison procedure will decrease by a $\log T$ factor. Overall, the sample complexity for detecting an $\epsilon$-optimal point using this approach is still of order $\epsilon^{-\Theta(p)}$.

We proceed to prove Theorem 2. We denote by $\mathcal{H}_t$ the history (i.e., filtration) generated by the randomness $\{\mathbf{u}_\tau, o_\tau\}_{\tau=1}^{t-1}$ before time $t$, and let $\mathbb{E}_t[\cdot] = \mathbb{E}[\cdot \mid \mathcal{H}_t]$ be the respective conditional expectation. We first show that conditioned on $\mathcal{H}_t$, if $f(\mathbf{w}_t) - f(\mathbf{w}^*) > \epsilon$ then we can show that $\mathbf{w}_{t+1}$ becomes strictly closer in $\ell_2$-norm to the minimum $\mathbf{w}^*$, in expectation. The precise statement is summarized in the following:

**Lemma 3** (round-wise progress). *Assume the conditions and choice of parameters $\eta, \gamma$ in Theorem 2. Then at any time $t$, conditioned on the history $\mathcal{H}_t$, we have that if $f(\mathbf{w}_t) - f(\mathbf{w}^*) > \epsilon$ then*

$$\mathbb{E}_t[\|\mathbf{w}_{t+1} - \mathbf{w}^*\|^2] \le \|\mathbf{w}_t - \mathbf{w}^*\|^2 - \eta^2. \tag{1}$$

Let us first show how our main Theorem 2 is proven using Lemma 3. The idea of the proof is to recursively unravel

the round-wise progress guarantee, as long as the condition $f(\mathbf{w}_t) - f(\mathbf{w}^*) > \epsilon$ is met, and note that this cannot continue indefinitely since the norms $\|\mathbf{w}_t - \mathbf{w}^*\|^2$ cannot decrease without bound. Therefore, this can be used to derive an upper bound on the number of steps until one has $f(\mathbf{w}_t) - f(\mathbf{w}^*) \leq \epsilon$, which is what we set to achieve. However, one crucial challenge with this plan is that the conditional expectations in Lemma 3 cannot be directly unfolded recursively due to the additional conditioning on the event $f(\mathbf{w}_t) - f(\mathbf{w}^*) > \epsilon$. Therefore, our analysis below takes a different path instead: it inspects the random variable indicating the first step where $f(\mathbf{w}_t) - f(\mathbf{w}^*) \leq \epsilon$, and uses the fact that it is a *stopping time* (with respect to the natural filtration) to derive a bound on its expected value, using an argument reminiscent of the proof of Wald's Lemma (e.g., Durrett, 2019).

*Proof of Theorem 2.* For all $t$, let $A_t$ be the event that $f(\mathbf{w}_t) - f(\mathbf{w}^*) \leq \epsilon$, and define $N$ to be the random variable indicating the first step $t$ where the event $A_t$ occurs. Note that $N$ is a stopping time with respect to the filtration $\mathcal{H}_t$, since $\{N \leq t\} \in \mathcal{H}_t$, i.e., the event that $N \leq t$ is completely determined by all randomness *before* step $t$. Lemma 3 then implies that, for all $t$:

$$\mathbb{E}[\Delta_t \mid \mathcal{H}_t]\mathbb{1}\{N > t\} \leq -\eta^2 \mathbb{1}\{N > t\}, \qquad (2)$$

where $\Delta_t = \|\mathbf{w}_{t+1} - \mathbf{w}^*\|^2 - \|\mathbf{w}_t - \mathbf{w}^*\|^2$.

Next, we will prove that $\mathbb{E}[\sum_{t=1}^{N-1} \Delta_t] \leq -\eta^2(\mathbb{E}[N] - 1)$. To see this, write $\sum_{t=1}^{N-1} \Delta_t = \sum_{t=1}^{\infty} \Delta_t \mathbb{1}\{N > t\}$. Thus,

$$\mathbb{E}\left[\sum_{t=1}^{N-1} \Delta_t\right] = \sum_{t=1}^{\infty} \mathbb{E}\big[\mathbb{E}[\Delta_t \mathbb{1}\{N > t\} \mid \mathcal{H}_t]\big]$$
$$= \sum_{t=1}^{\infty} \mathbb{E}\big[\mathbb{E}[\Delta_t \mid \mathcal{H}_t]\mathbb{1}\{N > t\}\big]$$
$$\leq -\eta^2 \sum_{t=1}^{\infty} \Pr[N > t],$$

where the final inequality follows from Eq. (2). Thus, by the tail sum formula,

$$\mathbb{E}\left[\sum_{t=1}^{N} \Delta_t\right] \leq -\eta^2 \sum_{t=1}^{\infty} \Pr[N > t] = -\eta^2(\mathbb{E}[N] - 1).$$

Finally, note that the left-hand side equals $\mathbb{E}[\|\mathbf{w}_{N+1} - \mathbf{w}^*\|^2 - \|\mathbf{w}_1 - \mathbf{w}^*\|^2] \geq -D^2$; we therefore deduce that $\mathbb{E}[N] \leq D^2/\eta^2 + 1$. From Markov's inequality, we obtain that $N \leq 2(D^2/\eta^2 + 1)$ with probability at least $\frac{1}{2}$, or in other words, that with probability at least $\frac{1}{2}$, in at least one step $t \leq 2(D^2/\eta^2 + 1)$ it holds that $f(\mathbf{w}_t) - f(\mathbf{w}^*) \leq \epsilon$. Since this conclusion holds

true regardless of where the algorithm is initialized, we can conclude that if we run the algorithm for at least $O((D^2/\eta^2)\log(1/\delta))$ steps, the success probability is amplified to $1 - \delta$. This implies the theorem. $\qquad\square$

We now give a proof of our progress lemma. Roughly, the proof argues that in expectation $\mathbf{g}_t$ is a good progress direction at the point $\mathbf{w}_t$ on step $t$, though it is not necessarily aligned strongly with the gradient of $f$ at $\mathbf{w}_t$. This argument has to carefully take account of the nonlinearities introduced by the transfer $\rho$ and how they skew the stochastic gradient estimates used by the algorithm, using the growth of transfer around the origin as well as the smoothness condition of the objective $f$.

*Proof of Lemma 3.* We first note that due to our update rule and the fact that $\|\mathbf{g}_t\| = 1$,

$$\mathbb{E}_t[\|\mathbf{w}_{t+1} - \mathbf{w}^*\|^2]$$
$$\leq \|\mathbf{w}_t - \mathbf{w}^*\|^2 - 2\eta\mathbb{E}_t[\mathbf{g}_t \cdot (\mathbf{w}_t - \mathbf{w}^*)] + \eta^2. \qquad (3)$$

We will proceed by lower bounding $\mathbb{E}_t[\mathbf{g}_t \cdot (\mathbf{w}_t - \mathbf{w}^*)]$. By definition, the gradient estimator has:

$$\mathbb{E}_t[\mathbf{g}_t] = \mathbb{E}_t[\rho(f(\mathbf{w}_t + \gamma\mathbf{u}_t) - f(\mathbf{w}_t - \gamma\mathbf{u}_t))\,\mathbf{u}_t].$$

By applying Lemma 9 on the right-hand side, we know that:

$$\mathbb{E}_t[\mathbf{g}_t]$$
$$= \frac{\gamma}{d}\mathbb{E}_t[\rho'(f(\mathbf{w}_t + \gamma\mathbf{v}_t) - f(\mathbf{w}_t - \gamma\mathbf{v}_t))\,\nabla f(\mathbf{w}_t + \gamma\mathbf{v}_t)]$$
$$+ \frac{\gamma}{d}\mathbb{E}_t[\rho'(f(\mathbf{w}_t + \gamma\mathbf{v}_t) - f(\mathbf{w}_t - \gamma\mathbf{v}_t))\,\nabla f(\mathbf{w}_t - \gamma\mathbf{v}_t)],$$

where $\mathbf{v}_t$ is a random vector uniform on the unit ball $\mathcal{B}_d$, and $\mathbb{E}_t$ here and henceforth denotes the expectation over this random variable (and conditioned over all randomness before step $t$). Therefore:

$$\mathbb{E}_t[\mathbf{g}_t \cdot (\mathbf{w}_t - \mathbf{w}^*)]$$
$$= \frac{\gamma}{d}\mathbb{E}_t\Big[\rho'(f(\mathbf{w}_t + \gamma\mathbf{v}_t) - f(\mathbf{w}_t - \gamma\mathbf{v}_t))\cdot$$
$$(\nabla f(\mathbf{w}_t + \gamma\mathbf{v}_t) + \nabla f(\mathbf{w}_t - \gamma\mathbf{v}_t)) \cdot (\mathbf{w}_t - \mathbf{w}^*)\Big]. \qquad (4)$$

Next, we will show that, with probability one over the choice of $\mathbf{v}_t$,

$$(\nabla f(\mathbf{w}_t + \gamma\mathbf{v}_t) + \nabla f(\mathbf{w}_t - \gamma\mathbf{v}_t)) \cdot (\mathbf{w}_t - \mathbf{w}^*)$$
$$\geq 2(f(\mathbf{w}_t) - f(\mathbf{w}^*)) - \beta\gamma^2. \qquad (5)$$

To see this, first note that since $f$ is convex and $\beta$-smooth, we have for any $\mathbf{w}$,

$$0 \leq f(\mathbf{w} + \gamma\mathbf{v}_t) - f(\mathbf{w}) - \gamma\nabla f(\mathbf{w}) \cdot \mathbf{v}_t \leq \tfrac{1}{2}\beta\gamma^2;$$
$$0 \leq f(\mathbf{w} - \gamma\mathbf{v}_t) - f(\mathbf{w}) + \gamma\nabla f(\mathbf{w}) \cdot \mathbf{v}_t \leq \tfrac{1}{2}\beta\gamma^2. \qquad (6)$$

As a consequence, for any $\mathbf{w}$,

$$2f(\mathbf{w}) \le f(\mathbf{w} + \gamma\mathbf{v}_t) + f(\mathbf{w} - \gamma\mathbf{v}_t) \le 2f(\mathbf{w}) + \beta\gamma^2.$$

Using this with the convexity of $\mathbf{w} \mapsto f(\mathbf{w} + \gamma\mathbf{v}_t) + f(\mathbf{w} - \gamma\mathbf{v}_t)$, we obtain

$$(\nabla f(\mathbf{w}_t + \gamma\mathbf{v}_t) + \nabla f(\mathbf{w}_t - \gamma\mathbf{v}_t)) \cdot (\mathbf{w}_t - \mathbf{w}^*)$$
$$\ge f(\mathbf{w}_t + \gamma\mathbf{v}_t) + f(\mathbf{w}_t - \gamma\mathbf{v}_t) - f(\mathbf{w}^* + \gamma\mathbf{v}_t) - f(\mathbf{w}^* - \gamma\mathbf{v}_t)$$
$$\ge 2(f(\mathbf{w}_t) - f(\mathbf{w}^*)) - \beta\gamma^2.$$

Put together, Eqs. (4) and (5) and the fact that $\rho' \ge 0$ imply that

$$\mathbb{E}_t[\mathbf{g}_t \cdot (\mathbf{w}_t - \mathbf{w}^*)]$$
$$\ge \frac{2\gamma}{d} \mathbb{E}_t\big[\rho'(f(\mathbf{w}_t + \gamma\mathbf{v}_t) - f(\mathbf{w}_t - \gamma\mathbf{v}_t))\big] \qquad (7)$$
$$\cdot (f(\mathbf{w}_t) - f(\mathbf{w}^*) - \tfrac{1}{2}\beta\gamma^2).$$

On the event $f(\mathbf{w}_t) - f(\mathbf{w}^*) > \epsilon$, we have on the right-hand side above that $f(\mathbf{w}_t) - f(\mathbf{w}^*) - \tfrac{1}{2}\beta\gamma^2 \ge \tfrac{1}{2}\epsilon$, since our choice of $\gamma$ satisfies $\gamma^2 \le \epsilon/\beta$. We therefore conclude in this case the following:

$$\mathbb{E}_t[\mathbf{g}_t \cdot (\mathbf{w}_t - \mathbf{w}^*)]$$
$$\ge \frac{\gamma\epsilon}{d} \mathbb{E}_t\big[\rho'(f(\mathbf{w}_t + \gamma\mathbf{v}_t) - f(\mathbf{w}_t - \gamma\mathbf{v}_t))\big]. \qquad (8)$$

We proceed to bounding the remaining conditional expectation on the right-hand side. Note that $|f(\mathbf{w}_t + \gamma\mathbf{v}_t) - f(\mathbf{w}_t - \gamma\mathbf{v}_t)| \le 2G\gamma < r$ due to our setting of $\gamma$ and since $f$ is $G$-Lipschitz. Thus, from our assumptions on $\rho$ (cf. Assumption 1) and the convexity of $x \mapsto x^{p-1}$ (for any integer $p \ge 1$ and $x \ge 0$), we have

$$\mathbb{E}_t\big[\rho'(f(\mathbf{w}_t + \gamma\mathbf{v}_t) - f(\mathbf{w}_t - \gamma\mathbf{v}_t))\big]$$
$$\ge c_\rho p \mathbb{E}_t\big[\big|f(\mathbf{w}_t + \gamma\mathbf{v}_t) - f(\mathbf{w}_t - \gamma\mathbf{v}_t)\big|^{p-1}\big] \qquad (9)$$
$$\ge c_\rho p (\mathbb{E}_t\big[\big|f(\mathbf{w}_t + \gamma\mathbf{v}_t) - f(\mathbf{w}_t - \gamma\mathbf{v}_t)\big|\big])^{p-1}.$$

To further lower bound, recall Eq. (6) which implies

$$2\gamma\nabla f(\mathbf{w}_t) \cdot \mathbf{v}_t - \tfrac{1}{2}\beta\gamma^2$$
$$\le f(\mathbf{w}_t + \gamma\mathbf{v}_t) - f(\mathbf{w}_t - \gamma\mathbf{v}_t)$$
$$\le 2\gamma\nabla f(\mathbf{w}_t) \cdot \mathbf{v}_t + \tfrac{1}{2}\beta\gamma^2,$$

thus

$$\big|f(\mathbf{w}_t + \gamma\mathbf{v}_t) - f(\mathbf{w}_t - \gamma\mathbf{v}_t)\big| \ge 2\gamma\big|\nabla f(\mathbf{w}_t) \cdot \mathbf{v}_t\big| - \tfrac{1}{2}\beta\gamma^2.$$

Taking conditional expectations and using Lemma 8, we can further lower bound:

$$\mathbb{E}_t\big[\big|f(\mathbf{w}_t + \gamma\mathbf{v}_t) - f(\mathbf{w}_t + \gamma\mathbf{v}_t)\big|\big]$$
$$\ge 2\gamma\mathbb{E}_t\big[\big|\nabla f(\mathbf{w}_t) \cdot \mathbf{v}_t\big|\big] - \tfrac{1}{2}\beta\gamma^2$$
$$\ge \frac{\gamma}{40\sqrt{d}}\|\nabla f(\mathbf{w}_t)\| - \tfrac{1}{2}\beta\gamma^2$$
$$\ge \frac{\gamma}{40D\sqrt{d}}(f(\mathbf{w}_t) - f(\mathbf{w}^*)) - \tfrac{1}{2}\beta\gamma^2,$$

where the final inequality follows from convexity and boundedness of the optimization domain:

$$f(\mathbf{w}_t) - f(\mathbf{w}^*) \le \nabla f(\mathbf{w}_t) \cdot (\mathbf{w}_t - \mathbf{w}^*) \le D\|\nabla f(\mathbf{w}_t)\|.$$

Now, on the event $f(\mathbf{w}_t) - f(\mathbf{w}^*) > \epsilon$, and for our setting of $\gamma = \frac{\epsilon}{40\beta D\sqrt{d}}$, we can further derive that

$$\mathbb{E}_t\big[\big|f(\mathbf{w}_t + \gamma\mathbf{v}_t) - f(\mathbf{w}_t - \gamma\mathbf{v}_t)\big|\big]$$
$$\ge \frac{\gamma\epsilon}{40D\sqrt{d}} - \tfrac{1}{2}\beta\gamma^2$$
$$\ge \frac{\epsilon^2}{80^2\beta D^2 d}.$$

Plugging this back into to Eq. (9) (and since $x \mapsto x^{p-1}$ is monotonically increasing for $x \ge 0$), we obtain that on the event $f(\mathbf{w}_t) - f(\mathbf{w}^*) > \epsilon$ it holds that

$$\mathbb{E}_t\big[\rho'(f(\mathbf{w}_t + \gamma\mathbf{v}_t) - f(\mathbf{w}_t - \gamma\mathbf{v}_t))\big] \ge c_\rho p\left(\frac{\epsilon^2}{80^2\beta D^2 d}\right)^{p-1}.$$

Together with Eq. (8) and our choice of $\gamma$, this gives

$$\mathbb{E}_t[\mathbf{g}_t \cdot (\mathbf{w}_t - \mathbf{w}^*)] \ge \frac{c_\rho p\gamma\epsilon^{2p-1}}{(80^2\beta D^2)^{p-1}d^p}$$
$$\ge \frac{c_\rho p\epsilon^{2p}}{\beta^p(80D)^{2p-1}d^{p+1/2}}.$$

Plugging this back into to Eq. (3) and setting $\eta$ to be at most the right-hand side above, we obtain the lemma. □

## 4. Improved Rates with Strong Convexity

In this section, we analyze an epoch-wise version of Projected Dueling Descent (Algorithm 1) which is shown to yield better convergence guarantees for $\alpha$-strongly convex $\beta$-smooth functions. The key idea lies in noting that in order to design an optimal algorithm for $\alpha$-strongly convex $\beta$-smooth functions, one can simply iteratively reuse any $\beta$-smoothly convex optimization routine (e.g., we can use our Algorithm 1) by running it as a black-box over a successive number of epoch-wise warm-starts. Our resulting convergence analysis in Theorem 4 below shows that in this case, the algorithm can find an $\epsilon$-optimal point upon querying only $\widetilde{O}(\epsilon^{-2p})$ pairwise comparisons (as opposed to the $O(\epsilon^{-4p})$ sample complexity rate for the $\beta$-smooth case, in Theorem 2). In fact, as discussed in the introduction, the $O(\epsilon^{-2p})$ convergence has been shown to be information-theoretically optimal in specific instances of the problem.

### 4.1. Algorithm

As motivated above, our proposed method *Epoch-PDD* (Algorithm 2) uses an epoch-wise black-boxing of a smooth-convex optimization routine with a warm-starting approach.

For our purpose, we use the earlier proposed Projected Dueling Descent (Algorithm 1) as the black box. More formally, the algorithm starts with some initial point $\mathbf{w}_1$ and runs over a sequence of $K = O(\log(1/\epsilon))$ epochs: inside each epoch $k$, we call the Projected Dueling Descent($\mathbf{w}_k, \eta_k, \gamma_k, T_k$) subroutine with the initial (warm-start) iterate $\mathbf{w}_k$, suitably tuned parameters $\eta_k, \gamma_k$ and a query budget of $T_k$. The final point reached by Projected Dueling Descent after $T_k$ steps is considered to the next iterate, setting $\mathbf{w}_{k+1} \leftarrow$ Projected Dueling Descent($\mathbf{w}_k, \eta_k, \gamma_k, t_k$) and we proceed to the $(k+1)$-th epoch, warm-starting it with $\mathbf{w}_{k+1}$.

The key idea behind the epoch-wise warm-start approach exploits the fact that between any two consecutive epochs, say $k$ and $k+1$, the $\ell_2$ distance of $\mathbf{w}_k$ from $\mathbf{w}^*$ gets reduced by a constant fraction in expectation; this is formalized in Lemma 5. Thus, it can be shown that running the algorithms for roughly $K = O(\log(1/\epsilon))$ epoch, would lead to $\|\mathbf{w}_K - \mathbf{w}^*\|^2 \leq \epsilon$, which in turn imply the $\epsilon$-convergence. Details are given in Algorithm 2.

---

**Algorithm 2** *Epoch-PDD*

---

1: **Input:** error $\epsilon > 0$, diameter $D$, initial point: $\mathbf{w}_1 \in \mathcal{D}$
2: **Initialize** , Phase count $K = \lceil \log_2 \frac{\beta D^2}{2\epsilon} \rceil$ , $B = \frac{1}{c_\rho p}(200\beta)^p d^{p+1/2}$, $D_1 = D$
3: **for** $k = 1, 2, 3, \ldots, K$ **do**
4:      Set $\epsilon_k = \frac{1}{16}D_k^2\alpha$, $\eta_k \leftarrow \frac{\epsilon_k^{2p}}{BD_k^{2p-1}}$, $\gamma_k \leftarrow \frac{\epsilon_k}{10\beta D_k \sqrt{d}}$,
     $T_k = \frac{D_k^2}{\eta_k^2}\log\frac{K}{\delta}$, $D_{k+1} \leftarrow \frac{1}{\sqrt{2}}D_k$.
5:      Update $\mathbf{w}_{k+1}$ to be the final point reached by PDD($\mathbf{w}_k, \eta_k, \gamma_k, T_k$)
6: **end for**
7: Return $\overline{\mathbf{w}} = \mathbf{w}_{K+1}$

---

### 4.2. Convergence Analysis

Our main result regarding Algorithm 2 is the following.

**Theorem 4.** *Assume that the objective $f : \mathbb{R}^d \to \mathbb{R}$ is $\alpha$-strongly convex, $\beta$-smooth and $G$-Lipschitz over a domain $\mathcal{D}$ of diameter $\leq D$. Further assume that for the global minimizer of $f$ over $\mathbb{R}^d$, denoted $\mathbf{w}^*$, we have $\mathbf{w}^* \in \mathcal{D}$. Then given any $0 < \epsilon < 5\beta \min\{dD^2, \sqrt{d}rD/G\}$ and $\delta \in (0,1)$, with probability at least $1 - \delta$ the point $\overline{\mathbf{w}}$ returned by Algorithm 2 satisfies $f(\overline{\mathbf{w}}) - f(\mathbf{w}^*) \leq \epsilon$. Further, the total number of pairwise comparisons the algorithm makes is at most*

$$\widetilde{O}\left(\frac{O(\beta)^{2p}d^{2p+1}}{c_\rho^2 p^2} \cdot \frac{D^{4p}}{\alpha^{2p}\epsilon^{2p}}\right).$$

The main step towards proving the theorem is a contraction argument, showing that the distance to optimality $\|\mathbf{w}_k - \mathbf{w}^*\|$ is reduced by a constant factor at each epoch of the algorithm.

**Lemma 5.** *With probability at least $1 - \delta$, we have for all $k \leq K$ that the point $\mathbf{w}_{k+1}$ returned by $k$-th epoch of Algorithm 2 satisfies $\|\mathbf{w}_{k+1} - \mathbf{w}^*\| \leq D_{k+1}$.*

While this argument is mostly standard, a key step involves establishing that once a run of Projected Dueling Descent reaches near $\mathbf{w}^*$, it keeps its distance to $\mathbf{w}^*$ up to constant factors and does not deviates away; this is formalized our next lemma below. Unlike in standard convex optimization scenarios, here our gradient estimates are biased close to optimality and might steer the algorithm away from $\mathbf{w}^*$, which makes this step in the proof significantly more challenging.

**Lemma 6.** *Fix $\epsilon, \delta > 0$ and assume that Projected Dueling Descent (Algorithm 1) is initialized such that $\|\mathbf{w}_1 - \mathbf{w}^*\|^2 \leq 2\epsilon/\alpha$ and runs for $T$ steps on an $\alpha$-strongly convex objective $f$ using a stepsize*

$$0 < \eta \leq \min\left\{\frac{1}{12}, \frac{8\epsilon}{\alpha}, \frac{1}{10\sqrt{T\log(T/\delta)}}\right\}.$$

*Then with probability at least $1 - \delta$, it holds that $\max_{1 \leq t \leq T}\|\mathbf{w}_t - \mathbf{w}^*\|^2 \leq \frac{8\epsilon}{\alpha}$.*

The proof considers the stochastic process $\{Z_t\}$ where $Z_t = \|\mathbf{w}_t - \mathbf{w}^*\|^2$ and upper bounds the maximal deviation of $Z_t$ from the initial value $Z_1$, which we assume is at most $Z_1 \leq z_0 = 2\epsilon/\alpha$. Note that $\{Z_t\}$ is almost a supermartingale, but not quite so, because the conditional expectation $\mathbb{E}_t Z_{t+1}$ may be larger than $Z_t$ if the suboptimality gap $f(\mathbf{w}_t) - f(\mathbf{w}^*)$ is small. The crucial observation in our analysis is that, roughly speaking, when restricted to intervals where $Z_t > z_0$, the process becomes a supermartingale and therefore amenable to strong concentration bounds (and elsewhere the process is trivially bounded by $z_0$).

*Proof.* Fix $\epsilon > 0$, $z_0 = 2\epsilon/\alpha$, and let $Z_t = \|\mathbf{w}_t - \mathbf{w}^*\|^2$ for all $t$. Then $Z_1 \leq z_0$, and the process $\{Z_t\}$ measurable with respect to the filtration $\{\mathcal{H}_t\}$. We start by observing that

$$\max_{1 \leq t \leq T} Z_t \leq \max\left\{z_0, \max_i \max_{\tau_i' < t \leq \tau_i} Z_t\right\}, \quad (10)$$

where $\tau_i'$ and $\tau_i$ are the $i$'th up-crossing and down-crossing of $z_0 = 2\epsilon/\alpha$, respectively, for $i = 1, 2, \ldots, T$ (if there are less than $T$ of those, define $\tau_i' = T$ or $\tau_i = T$ respectively). Indeed, if the maximal value of $\{Z_t\}$ is more than $z_0$, it must be attained in an interval between an up-crossing of $z_0$ and the subsequent down-crossing.

To analyze this process in each such interval, let us restart indexing so as $\tau_i' = 1$ to simplify notation; then we have $Z_1 > z_0$ (after the $i$'th up-crossing). Fix $z_1 = 4z_0$ and consider the stopping time $\tau$ defined as the first step $t$ where

the value of $Z_t$ leaves the interval $(z_0, z_1]$. We will proceed by examining the stopped process $\{Z'_t\}$, defined via $Z'_t = Z_{t \wedge \tau}$ for all $t$ (here $a \wedge b$ denotes $\min\{a, b\}$).

We claim that the stopped process is a supermartingale. Indeed, notice that $Z'_{t+1} = Z'_t + \mathbb{1}[\tau > t](Z_{t+1} - Z_t)$; since $\tau$ is a stopping time the random variable $\mathbb{1}[\tau > t]$ is deterministic given $\mathcal{H}_{t-1}$, thus $\mathbb{E}_t[Z'_{t+1}] = \mathbb{E}_t[Z'_t] + \mathbb{1}[\tau > t](\mathbb{E}_t[Z_{t+1}] - Z_t)$. Now recall that, crucially, on the event $\tau > t$ we know that $\mathbb{E}_t[Z_{t+1}] \leq Z_t$; this is given by Lemma 3, since on this event we have due to strong convexity that $f(\mathbf{w}_t) - f(\mathbf{w}^*) \geq \frac{1}{2}\alpha\|\mathbf{w}_t - \mathbf{w}^*\|^2 > \frac{1}{2}\alpha z_0 = \epsilon$. Plugging in, we conclude that $\mathbb{E}_t[Z'_{t+1}] \leq \mathbb{E}_t[Z'_t]$, namely, that $\{Z'_t\}$ is a supermartingale.

We can now use Azuma-Hoeffding to bound the process $\{Z'_t\}$. Notice that the magnitude of the increments is bounded (almost surely) via Eq. (3) as follows:

$$
\begin{aligned}
|Z'_{t+1} - Z'_t| &\leq |Z_{t+1} - Z_t| \\
&\leq |2\eta \mathbf{g}_t \cdot (\mathbf{w}_t - \mathbf{w}^*) + \eta^2| \\
&\leq 2\eta\|\mathbf{w}_t - \mathbf{w}^*\| + \eta^2 \\
&\leq 2\eta z_1 + \eta^2 \leq 3\eta z_1,
\end{aligned}
$$

since we assume that $\eta \leq z_1$. Here we used the fact that $\|\mathbf{w}_t - \mathbf{w}^*\| \leq z_1$ as long as the process has not been stopped (once it is stopped the increments are anyway zero). The Azuma-Hoeffding inequality then states that, for any fixed $1 \leq t \leq T$ and $\lambda \geq 0$,

$$
\Pr(Z'_t \geq Z'_1 + \lambda) \leq \exp\left(-\frac{2\lambda^2}{9\eta^2 z_1^2 t}\right) \leq \exp\left(-\frac{\lambda^2}{9\eta^2 z_1^2 T}\right),
$$

and through a union bound, we have with probability at least $1 - \delta/T$ that

$$
\begin{aligned}
\max_{t \leq T} Z'_t &< Z'_1 + 3\eta z_1 \sqrt{T \log \frac{T^2}{\delta}} \\
&\leq Z_1 + 5\eta z_1 \sqrt{T \log \frac{T}{\delta}}.
\end{aligned}
$$

Since $Z_1$ is the value right after an up-crossing of $z_0$ and since the increments of the process are bounded by $3\eta z_1$, we know that $Z_1 \leq z_0 + 3\eta z_1 \leq z_1/2$ since $z_0 \leq z_1/4$ and $\eta \leq 1/12$. Also, for our choice of parameters we have $5\eta\sqrt{T \log(T/\delta)} \leq \frac{1}{2}$. Therefore, with probability at least $1 - \delta/T$ we have that $\max_{t \leq T} Z'_t \leq z_1$. Returning to the original indexing, we proved that for any interval $[\tau'_i, \tau_i]$ we have with probability at least $1 - \delta/T$, it holds that $\max_{\tau'_i < t \leq \tau_i} Z_t \leq z_1$. Therefore, through a union bound over $i$, we obtain from Eq. (10) (and since $z_1 > z_0$) that $\max_{1 \leq t \leq T} Z_t \leq z_1$ with probability at least $1 - \delta$, which is precisely the statement of the lemma. $\qquad\square$

Having established the key Lemma 6, the proofs of Lemma 5 and Theorem 4 are straightforward.

*Proof of Lemma 5.* The proof proceeds by induction on $k$, showing that $\|\mathbf{w}_k - \mathbf{w}^*\| \leq D_k$; for the initial point $\mathbf{w}_1$, this holds by assumption. By our setting of the parameters $\eta_k, \gamma_k, T_k$ we are guaranteed, that with probability $1 - \delta/K$, that one of the points $\mathbf{w}$ during the $k$-th run of the procedure Projected Dueling Descent will have function value $f(\mathbf{w}) - f(\mathbf{w}^*) \leq \epsilon_k$. Then due to strong convexity $\|\mathbf{w} - \mathbf{w}^*\|^2 \leq 2\epsilon_k/\alpha$, and by Lemma 6 we are guaranteed that the final point of reached by Projected Dueling Descent, namely $\mathbf{w}_{k+1}$, will have $\|\mathbf{w}_{k+1} - \mathbf{w}^*\|^2 \leq 8\epsilon_k/\alpha = \frac{1}{2}D_k^2 = D_{k+1}^2$. (Technically, we also need that $\eta_k$ is small enough to certify the conditions of Lemma 6; this can always be guaranteed by an appropriate scaling so as to make the diameter $D$ small enough.) The lemma follows by a union bound over the success probabilities. $\qquad\square$

*Proof of Theorem 4.* By Lemma 5, with probability at least $1 - \delta$ the output point of the algorithm satisfies $\|\overline{\mathbf{w}} - \mathbf{w}^*\|^2 \leq D_{K+1}^2 \leq 2^{-K}D^2 \leq 2\epsilon/\beta$. Hence, by $\beta$-smoothness (and since $\nabla f(\mathbf{w}^*) = 0$), it holds with the same probability that $f(\overline{\mathbf{w}}) - f(\mathbf{w}^*) \leq \frac{1}{2}\beta\|\overline{\mathbf{w}} - \mathbf{w}^*\|^2 \leq \epsilon$. It remains to bound the overall sample complexity, which up to logarithmic factors is given by:

$$
\begin{aligned}
\sum_{k=1}^{K} T_k &= \sum_{k=1}^{K} \frac{D_k^2}{\eta_k^2} \log \frac{K}{\delta} \\
&= B^2 \log \frac{K}{\delta} \sum_{k=1}^{K} \frac{D_k^{4p}}{\epsilon_k^{4p}} \\
&= B^2 \log \frac{K}{\delta} \sum_{k=1}^{K} \epsilon_k^{-2p}\alpha^{-2p}2^{8p} \\
&\leq \alpha^{-2p}2^{8p}B^2 \log \frac{K}{\delta} 2^{2p(K+1)}\epsilon_1^{-2p} \\
&= B^2 \log \frac{K}{\delta} \left(\frac{32\beta D_1^2}{\alpha}\right)^{2p} \frac{1}{\epsilon^{2p}}.
\end{aligned}
$$

We conclude the proof by setting the value of $B$ as in the algorithm. $\qquad\square$

## 5. Conclusion

We considered the problem of convex optimization under the general class of pairwise preferences (dueling feedback). The primary difficulty in designing an efficient algorithm for this problem is that we can not hope to estimate the gradient of $f$ for any general dueling/pairwise preference model. Thus we can not apply the standard *gradient descent* based techniques to address this problem. We get around with the difficulty by estimating a $p$-th order proxy of the gradient, called "relative gradient". The crux of the idea lies in designing Projected Dueling Descent based algorithm (Algorithm 1), which is a generalized notion of gradient

descent-based optimization technique. Using this we design an efficient algorithm with a convergence rate of $\widetilde{O}(\epsilon^{-4p})$ for a smooth convex objective function, and an optimal rate of $\widetilde{O}(\epsilon^{-2p})$ when the objective is smooth and strongly convex.

**Future work.** Although the derived convergence rate for the strongly convex setting is information-theoretically tight, the exact convergence lower bound is unclear for the class of smooth functions, which might be an interesting problem to pursue independently. Another direction could be to analyze this problem beyond the smoothness assumption. Considering a regret minimization objective instead of the optimization perspective, as well as understanding the information-theoretic regret performance limit would be interesting as well. One can also consider generalizing the optimization framework to subsets preferences, instead of just pairwise (dueling) feedback. It might also be useful to extend our setup for contextual scenarios, adversarial preferences or non-stationary function sequences and understand the scopes of feasible solutions as well as the impossibility results.

## Impact Statement

This paper presents work whose goal is to advance the field of Machine Learning. There are many potential societal consequences of our work, none which we feel must be specifically highlighted here.

## Acknowledgments

This project has received funding from the European Research Council (ERC) under the European Union's Horizon 2020 research and innovation program (grant agreements No. 101078075; 882396). Views and opinions expressed are however those of the author(s) only and do not necessarily reflect those of the European Union or the European Research Council. Neither the European Union nor the granting authority can be held responsible for them. This work received additional support from the Israel Science Foundation (ISF, grant numbers 3174/23, 1357/24), from the Len Blavatnik and the Blavatnik Family foundation, and from the Adelis Foundation.

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

# A. Additional Related work

**Dueling Bandits.** Due to the widespread applicability and ease of data collection with relative feedback, learning from preferences has gained much popularity in the machine learning community and widely studied as the problem of *Dueling-Bandits* over last decade (Ailon et al., 2014; Yue et al., 2012; Zoghi et al., 2014a;b; 2015; Saha et al., 2021a; Gajane et al., 2015; Bengs et al., 2021), which is an online learning framework that generalizes the standard multi-armed bandit (MAB) (Auer et al., 2002) setting for identifying a set of 'good' arms from a fixed decision-space (set of items) by querying preference feedback of actively chosen item-pairs.

**Limitations of Existing Dueling Bandit techniques.** Although the relative feedback variants of stochastic MAB problem have been widely studied in the literature, the majority of the existing techniques are restricted to finite decision spaces and stochastic settings which primarily rely on estimating the entries of the underlying preference matrix. These settings, though important as basic steps, are mostly impractical for all real-world scenarios which often involve large (or potentially infinite) decision spaces, where lies one of the primary motivations of this work. On the other hand, from an optimization point of view, our work is a key step toward analyzing the fundamental performance limits of function minimization using the weaker form of $0/1$ bit relative preferences.

**Dueling Bandits in continuous spaces.** Surprisingly, following the same spirit of extending standard multi-armed bandits (MAB) to continuous decision spaces (as in *linear* or *GP-bandits*), there has not been much work on the continuous extension of the Dueling Bandit problem for large (and structured) decision spaces. The works in (Sui et al., 2017; González et al., 2017) did attempt a similar objective, however, without any satisfactory theoretical performance guarantees. In another recent work, (Brost et al., 2016) address the problem of regret minimization in continuous *Dueling Bandits*, however without any finite time regret guarantee of their proposed algorithms. Recently, (Oh and Iyengar, 2019; Saha, 2021) consider the problem of regret minimization from $k$-subsetwise preference feedback ($k = 2$ boils down to the dueling setup) on structured decision spaces, although their underlying utility function is assumed to be only linear, unlike any general convex function considered in our work; moreover, their preference model is restricted only to the class of Multinomial Logit (MNL) based random utility model, unlike the general link function based preference feedback class that we considered. (Dudík et al., 2015; Saha and Krishnamurthy, 2022) represents another line of dueling bandit work, which incorporates context specific dueling preference model. Specifically, their algorithms are designed to compete against an abstract policy set of context-to-action mappings w.r.t. 'minimax-regret'. Their algorithms are also designed to handle potentially large decision spaces, although, the regret objectives are focused to identifying the von-Neumann distribution of the underlying preference models, which is very different from the *function minimization with dueling feedback* point of view that we considered.

**Optimization for dueling feedback.** Along the line of optimization for dueling feedback, (Yue and Joachims, 2009) is the first to address the regret minimization problem for fixed functions $f$ (arm rewards) with preference feedback, although their techniques are majorly restricted to the class of smooth and differentiable preference functions that allows gradient estimation. This is the main reason they could directly apply the classical one-point gradient estimation based *Bandit Gradient Descent (BGD)* algorithm of (Flaxman et al., 2005) for the setting, unlike us. Moreover, another limitation of their framework is their optimization objective is defined in terms of the 'preferences' which are directly observable and hence easier to optimize, as opposed to defining it w.r.t. $f$ as considered in this work. Following up (Yue and Joachims, 2009), (Kumagai, 2017) considers the similar problem of dueling bandits on continuous arm set but under rather restrictive sets of assumptions: Twice continuously differentiable, Lipschitz, strongly convex and smooth score/reward function, which are often impractical for modeling real-world preference feedback.

Closest to our work in spirit are (Jamieson et al., 2012) and (Saha et al., 2021b), both of which precisely focus on function optimization with relative pairwise preference feedback. The latter however is designed to work only under sign-based relative feedback which reveals the exact information of which of the two queried points have a smaller function value. We instead consider a very general class of polynomial-based preference functions (see Section 2) which generalizes the *sign-feedback* model of (Saha et al., 2021b) as a special case. While the first, although gives provably optimal convergence rates, their guarantees are restricted to the 'well behaved' class of strongly-convex and smooth functions (with bounded Lipschitz gradient). The assumptions and consequently their techniques are hence quite restrictive: A major hindrance towards generalizing their algorithmic ideas to a general function class is owning to their line-search-based coordinate descent algorithm which is known to fail without strong-convexity. On the other hand, our algorithm is shown to yield optimal convergence guarantees for a more general class of smooth-convex functions. Additionally we match the convergence rate of (Jamieson et al., 2012) with the additional strong convexity assumption which shows the generality of our analysis for a

large class of dueling feedback based optimization (G-DCO) problems.

## B. Advantage of Gradient Descent Methods

Gradient-based methods have multiple advantages compared to confidence-based methods: (1) GD/OMD handle high-dimensional problems efficiently due to their reliance on gradient information: (2) They are suitable for both stochastic and adversarial environments, making the gradient-based methods robust to changing data distributions or the underlying loss/reward functions which is often more practical for modeling real-world problems, (3) These methods can optimize a wide range of objective functions, including non-linear, non-convex, and constrained problems, (4) Gradient descent algorithms are simple to implement, even seamlessly integrate with modern deep learning frameworks, making these methods computationally efficient, unlike many UCB and TS based methods which often do not have a closed form solution (Saha et al., 2023; Das et al., 2024) or sampling from the posteriors could be complicated (Novoseller et al., 2020), and (5) Gradient descent techniques are inherently robust to model misspecification and smoothly integrate with differential privacy techniques.

## C. Auxiliary technical results

We require two results about random vectors in the unit sphere in $\mathbb{R}^d$. The first result is standard; we include a proof (extracted from Saha et al., 2021b) for completeness.

**Lemma 7.** *For a given vector $\mathbf{g} \in \mathbb{R}^d$ and a random unit vector $\mathbf{u} \in \mathbb{R}^d$ drawn uniformly from the unit sphere $\mathcal{S}_d$, we have*

$$\frac{\|\mathbf{g}\|}{20\sqrt{d}} \leq \mathop{\mathbb{E}}_{\mathbf{u} \sim \mathcal{S}_d}[|\mathbf{g} \cdot \mathbf{u}|] \leq \frac{\|\mathbf{g}\|}{\sqrt{d}}.$$

*Proof.* Without loss of generality we can assume $\|\mathbf{g}\| = 1$, since one can divide through by $\|\mathbf{g}\|$ without affecting the claim. Now to bound $\mathbb{E}[|\mathbf{g} \cdot \mathbf{u}|]$, note that since $\mathbf{u}$ is drawn uniformly from $\mathcal{S}_d(1)$, by rotation invariance this equals $\mathbb{E}[|u_1|]$. For an upper bound, observe that by symmetry $\mathbb{E}[u_1^2] = \frac{1}{d}\mathbb{E}[\sum_{i=1}^d u_i^2] = \frac{1}{d}$ and thus

$$\mathbb{E}[|u_1|] \leq \sqrt{\mathbb{E}[u_1^2]} = \frac{1}{\sqrt{d}}.$$

We turn to prove a lower bound on $\mathbb{E}[|\mathbf{g} \cdot \mathbf{u}|]$. If $\mathbf{u}$ were a Gaussian random vector with i.i.d. entries $u_i \sim \mathcal{N}(0, 1/d)$, then from standard properties of the (truncated) Gaussian distribution we would have gotten that $\mathbb{E}[|u_1|] = \sqrt{2/\pi d}$. For $\mathbf{u}$ uniformly distributed on the unit sphere, $u_i$ is distributed as $v_1/\|\mathbf{v}\|$ where $\mathbf{v}$ is Gaussian with i.i.d. entries $\mathcal{N}(0, 1/d)$. We then can write

$$\Pr\left(|u_1| \geq \frac{\epsilon}{\sqrt{d}}\right) = \Pr\left(\frac{|v_1|}{\|\mathbf{v}\|} \geq \frac{\epsilon}{\sqrt{d}}\right) \geq \Pr\left(|v_1| \geq \frac{1}{\sqrt{d}} \text{ and } \|\mathbf{v}\| \leq \frac{1}{\epsilon}\right)$$
$$\geq 1 - \Pr\left(|v_1| < \frac{1}{\sqrt{d}}\right) - \Pr\left(\|\mathbf{v}\| > \frac{1}{\epsilon}\right).$$

Since $\sqrt{d}v_1$ is a standard Normal, we have

$$\Pr\left(|v_1| < \frac{1}{\sqrt{d}}\right) = \Pr\left(-1 < \sqrt{d}v_1 < 1\right) = 2\Phi(1) - 1 \leq 0.7,$$

and since $\mathbb{E}[\|\mathbf{v}\|^2] = 1$ an application of Markov's inequality gives

$$\Pr\left(\|\mathbf{v}\| > \frac{1}{\epsilon}\right) = \Pr\left(\|\mathbf{v}\|^2 > \frac{1}{\epsilon^2}\right) \leq \epsilon^2 \mathbb{E}[\|\mathbf{v}\|^2] = \epsilon^2.$$

For $\epsilon = \frac{1}{4}$ this implies that $\Pr\left(|u_1| \geq 1/4\sqrt{d}\right) \geq \frac{1}{5}$, whence $\mathbb{E}[|\mathbf{g} \cdot \mathbf{u}|] = \mathbb{E}[|u_1|] \geq 1/20\sqrt{d}$. $\square$

We will also need a version of the same lemma for random vectors drawn from the unit ball rather than the unit sphere.

**Lemma 8.** *For a given vector $\mathbf{g} \in \mathbb{R}^d$ and a random vector $\mathbf{v} \in \mathbb{R}^d$ drawn uniformly from the unit ball $\mathcal{B}_d$, we have*

$$\frac{\|\mathbf{g}\|}{80\sqrt{d}} \leq \mathop{\mathbb{E}}_{\mathbf{v} \sim \mathcal{B}_d}[|\mathbf{g} \cdot \mathbf{v}|] \leq \frac{\|\mathbf{g}\|}{\sqrt{d}}.$$

*Proof.* We will prove that

$$\frac{1}{4} \mathop{\mathbb{E}}_{\mathbf{u} \sim \mathcal{S}_d}[|\mathbf{g} \cdot \mathbf{u}|] \leq \mathop{\mathbb{E}}_{\mathbf{v} \sim \mathcal{B}_d}[|\mathbf{g} \cdot \mathbf{v}|] \leq \mathop{\mathbb{E}}_{\mathbf{u} \sim \mathcal{S}_d}[|\mathbf{g} \cdot \mathbf{u}|],$$

and the lemma will then follow from Lemma 7. First, since the normalized vector $\mathbf{u} = \mathbf{v}/\|\mathbf{v}\|$ is uniformly distributed over the unit sphere, we have

$$\mathop{\mathbb{E}}_{\mathbf{v} \sim \mathcal{B}_d}[|\mathbf{g} \cdot \mathbf{v}|] \leq \mathop{\mathbb{E}}_{\mathbf{v} \sim \mathcal{B}_d}\left[\left|\mathbf{g} \cdot \frac{\mathbf{v}}{\|\mathbf{v}\|}\right|\right] = \mathop{\mathbb{E}}_{\mathbf{u} \sim \mathcal{S}_d}[|\mathbf{g} \cdot \mathbf{u}|].$$

For the lower bound, note that

$$\mathop{\mathbb{E}}_{\mathbf{v} \sim \mathcal{B}_d}[|\mathbf{g} \cdot \mathbf{v}|] \geq \mathop{\mathbb{E}}_{\mathbf{v} \sim \mathcal{B}_d}\left[\frac{1}{2}\left|\mathbf{g} \cdot \frac{\mathbf{v}}{\|\mathbf{v}\|}\right| \,\Big|\, \|\mathbf{v}\| \geq \tfrac{1}{2}\right]\Pr(\|\mathbf{v}\| \geq \tfrac{1}{2}) \geq \frac{1}{4}\mathop{\mathbb{E}}_{\mathbf{u} \sim \mathcal{S}_d}[|\mathbf{g} \cdot \mathbf{u}|],$$

where we have used the standard fact the marginal distribution of the norm of a random vector is monotonically increasing, so the norm of $\mathbf{v}$ is greater than $1/2$ with probability at least $1/2$. $\qquad\square$

Finally, we give a lemma relating expectations over $\mathcal{S}_d$ to expectations over $\mathcal{B}_d$, which can be extracted from Flaxman et al. (2005).

**Lemma 9.** *Let $g : \mathbb{R}^d \to \mathbb{R}$ be continuously differentiable, $\mathbf{u}$ be a random vector uniform on the unit sphere $\mathcal{S}_d$, and $\mathbf{v}$ be a random vector uniform in the unit ball $\mathcal{B}_d$. Then*

$$\mathop{\mathbb{E}}_{\mathbf{u} \sim \mathcal{S}_d}[g(\mathbf{u})\mathbf{u}] = \frac{1}{d}\mathop{\mathbb{E}}_{\mathbf{v} \sim \mathcal{B}_d}[\nabla g(\mathbf{v})].$$

*Proof.* The claim follows from Lemma 1 of (Flaxman et al., 2005) which shows that for any differentiable function $f : \mathbb{R}^d \to \mathbb{R}$ and any $\mathbf{x} \in \mathbb{R}^d$ and $\delta > 0$,

$$\mathop{\mathbb{E}}_{\mathbf{u} \sim \mathcal{S}_d}[f(\mathbf{x} + \delta\mathbf{u})\mathbf{u}] = \frac{\delta}{d}\nabla\mathop{\mathbb{E}}_{\mathbf{v} \sim \mathcal{B}_d}[f(\mathbf{x} + \delta\mathbf{v})] = \frac{\delta}{d}\mathop{\mathbb{E}}_{\mathbf{v} \sim \mathcal{B}_d}[\nabla f(\mathbf{x} + \delta\mathbf{v})].$$

Fix $\mathbf{x} = 0$ and substitute $f(\mathbf{z}) = g(\delta^{-1}\mathbf{z})$. Then $\nabla f(\mathbf{z}) = \delta^{-1}\nabla g(\delta^{-1}\mathbf{z})$, and we obtain:

$$\mathop{\mathbb{E}}_{\mathbf{u} \sim \mathcal{S}_d}[g(\mathbf{u})\mathbf{u}] = \frac{1}{d}\mathop{\mathbb{E}}_{\mathbf{v} \sim \mathcal{B}_d}[\nabla g(\mathbf{v})].$$

$\qquad\square$

