# OpenReview forum: "Dueling Convex Optimization with General Preferences"
_ICML.cc/2025/Conference — ICML 2025 poster_

### Official Review · Reviewer_WhdB · 2025-03-11

**Overall Recommendation:** 3

**Summary:**

This paper considers the problem of dueling bandits with general preferences, where the preference model between two decisions (called the transfer function) is not specified to some specific choices. The most technical challenge in this problem is the estimation of gradients since only the preference feedback (a one-bit feedback generated from a Bernoulli distribution) is accessible, which is even harder than the one-point bandit feedback. To this end, the authors proposed a novel estimation of $g_t = o_t u_t$, which does not necessarily align with the true gradient but is still sufficient for a gradient descent update. As for results, the authors achieved an $O(\epsilon^{-4p})$ oracle query complexity to obtain an $\epsilon$-optimal decision. Furthermore, for strongly convex functions, the aforementioned bound can be further improved to $\tilde{O}(\epsilon^{-2p})$ using a similar idea like epoch-GD, which is theoretically optimal.

**Claims And Evidence:**

Yes.

**Essential References Not Discussed:**

No.

**Experimental Designs Or Analyses:**

There are no experiments in this paper.

**Methods And Evaluation Criteria:**

Yes.

**Other Comments Or Suggestions:**

I noticed that Thm 2 has an assumption of $0 < \epsilon < 5 \beta \min\\{dD^2, \sqrt{d} r D / G\\}$. What if the required error $\epsilon \ge \min\\{dD^2, \sqrt{d} r D / G\\}$? I guess this should not be a big deal since the problem would become easy if the required error is large.

**Other Strengths And Weaknesses:**

## Strengths
1. Studying general preference functions is a meaningful problem, which may be further applied to real-world applications, for example, RLHF.

2. The adopted assumption is mild and reasonable. The authors have provided detailed explanations to validate its rationality.

3. The proposed algorithm is simple and easy to implement. It generally follows the update rule of gradient descent but with a carefully chosen gradient estimation. The key challenge lies in the technical analysis of this update rule.

## Weaknesses
1. The guarantee of Theorem 2 only holds for some unknown decision generated in the optimization process. To find it out, additional operations are needed, which will lead to extra query complexity.

**Questions For Authors:**

Not many.

**Relation To Broader Scientific Literature:**

N/A.

**Theoretical Claims:**

Yes, mostly.

---

> ### Author Rebuttal · Authors · 2025-03-31
>
> > “The guarantee of Theorem 2 only holds for some unknown decision generated in the optimization process. To find it out, additional operations are needed, which will lead to extra query complexity.”
>
> You are right to notice that, due to the challenging feedback model, the nature of our convergence guarantee becomes more intricate than simply holding for the final iterate of the algorithm. We address this issue in detail in the paragraph in lines 167-177 (right column).
> For additional details regarding the $\log(T)$ factor see our reply to reviewer br4X.
>
> > “What if the required error $\epsilon \geq 5\beta\min(  d D^2 ,\sqrt{d}rD/G )$”
>
> Indeed, as you probably realized, in this case we can simply set $\epsilon=5\beta\min(  d D^2 ,\sqrt{d}rD/G )$ and obtain an only stronger guarantee, and the convergence bound will be independent of $\epsilon$ and it will only depend on the problem parameters $\beta,d,D,r,G$.

---

> > ### Comment · Reviewer_WhdB · 2025-04-07
> >
> > Thanks to the authors for the reply. I acknowledge the contributions of this work. However, as mentioned in the review comments, the condition on $\epsilon$ seems to be an assumption, which would confuse readers. As the authors have said, when the required error is larger, a stronger guarantee can be achieved. As a result, I highly recommend that the authors could add this case and its corresponding results to the theorem to make it self-contained. I maintain my current score.

---

> > > ### Author Response · Authors · 2025-04-07
> > >
> > > We appreciate the reviewer’s additional comments and thoughtful discussion.  We agree that a comment about the condition on $\epsilon$ is in order, to clarify that it does not limit the generality of our result in any way.  We will carefully incorporate this into the final version - thanks!

---

### Official Review · Reviewer_iszU · 2025-03-13

**Overall Recommendation:** 3

**Summary:**

This work studies convex optimization with dueling feedback and general transfer functions. The main contribution is an algorithm with $\epsilon^{-4p}$ convergence rate for smooth and convex functions and $\epsilon^{-2p}$ convergence rate for smooth and strongly-convex functions,  where $p$ is the minimal degree (with a non-zero coefficient) in the transfer’s series expansion about the origin.

**Claims And Evidence:**

The statements are supported with proofs.

**Essential References Not Discussed:**

NA

**Experimental Designs Or Analyses:**

NA

**Methods And Evaluation Criteria:**

NA

**Other Comments Or Suggestions:**

NA

**Other Strengths And Weaknesses:**

Strength:
This work improves the prior art by considering a very general class of transfer functions where $p$ can be greater than $1$. The rates in smooth and strongly convex objectives match the existing lower bounds.

Weakness:
However, this setting $p>1$ seems not very important. This work lacks discussions of meaningful examples and concrete problems/applications in practice.

**Questions For Authors:**

Are there any example of concrete problems in practice where $p > 1$?

**Relation To Broader Scientific Literature:**

NA

**Theoretical Claims:**

The proofs in the main paper seem to be correct.

---

> ### Author Rebuttal · Authors · 2025-03-31
>
> > “Are there any example of concrete problems in practice where p>1?”
>
> First, we should note that even for $p=1$ our results constitute the first convergence bounds for convex optimization with (approximately) linear transfer functions.
>
> Next, to your question, it is worth recalling what the transfer function $\rho$ abstracts. One motivation is similar to Reinforcement learning from human feedback (RLHF), where the transfer function abstracts how human preference behaves given similarly-valued alternatives. The most natural one is to assume that when the alternatives are of similar values, humans select almost randomly. The degree $p$ abstracts how small changes in the quality of the two alternatives are translated to differences in human evaluations. A higher $p$ value implies that differences are harder to detect. For this reason, $p=2$ is equally well-motivated as $p=1$, and there is no particular fundamental reason to model preferences using a linear function.
>
> Again, note that the transfer function transfers between the valuations (e.g. in LLM: how good is a response to a prompt) to the human response (e.g. in LLM: the likelihood that a human will prefer each alternative). There is no reason to believe the transfer should be necessarily linear.

---

### Official Review · Reviewer_br4X · 2025-03-14

**Overall Recommendation:** 3

**Summary:**

This paper proposes an algorithm for the setting of dueling convex optimization, with a broad class of transfer functions.

**Claims And Evidence:**

The claims are well supported.

**Essential References Not Discussed:**

N/A

**Experimental Designs Or Analyses:**

N/A

**Methods And Evaluation Criteria:**

The proposed methods make sense.

**Other Comments Or Suggestions:**

1, The concept of "admissible transfer functions" appears abruptly in the title of Subsection 2.2, without further explanations.
2. In terms of the structure of the paper, the main text includes too much proof detail and inadequate discussions of the main theorems.

**Other Strengths And Weaknesses:**

Strengths:
1. This work follows a clear logic, and is technically solid, especially the analysis about the stopping time.

Weaknesses:
1. This work could be improved by justifying the algorithm design with experiments.
2. In general, the paper lacks interpretation of the results. I was not sure whether the proposed algorithm can be compared against some naive algorithms, given that the algorithm is the first one for the setting being studied. The choice of hyperparameters also need more discussions.
3. The algorithm design of this paper is mostly an extension of Saha et al. (2021b). The authors could have over-claimed the contribution of the algorithm design, especially the "relative gradient".

**Questions For Authors:**

1. Could the authors further elaborate the following statement to solve the problem of unknown time index $t$ that achieves good performance: *Since errors accumulate only along the path from the root to the true minimum, both the accuracy and confidence of the entire comparison procedure will decrease by a $\log T$ factor.*
2. Can the assumption of $\rho(0)=0$ be relaxed? Can the results be improved if $\rho$ is an odd function?

**Relation To Broader Scientific Literature:**

The two most related works are Jamieson et al. (2012) and Saha et al. (2021). This work is a natural extension of Saha et al. (2021).

**Theoretical Claims:**

I checked the proof for the non-strongly-convex case.

---

> ### Author Rebuttal · Authors · 2025-03-31
>
> > “This work could be improved by justifying the algorithm design with experiments”
>
> This is primarily a theoretical work in optimization with partial feedback, and our main goal is to understand the fundamental achievable convergence bounds in this setting. This is also the focus of much of the prior work in this space. We feel that the provided theory gives ample justification for the algorithm.
>
> > “I was not sure whether the proposed algorithm can be compared against some naive algorithms, given that the algorithm is the first one for the setting being studied”
>
> When we have dueling feedback, it is even challenging to design “naive” algorithms for the general convex (and smooth) case. One could view the existing algorithm of Jamieson et al (2012) as an attempt to generalize a natural noisy binary search algorithm to a high-dimensional setting, but unfortunately, their convergence requires strong assumptions such as strong convexity as their extension relies on a coordinate descent scheme. Our main contribution is to give precise convergence bounds for the general convex (and smooth) dueling setting.
>
> > “The choice of hyperparameters also need more discussions”
>
> The hyperparameters of our algorithms are set to guarantee the best performance bound we can derive. The rationale for their choice is to optimize, in hindsight, the convergence bound we establish.
>
> > “The algorithm design of this paper is mostly an extension of Saha et al. (2021b)”
>
> First, we should emphasize that, in our view, expanding the scope of comparison-based / dueling optimization to a variety of nonlinear transfer functions is novel and significant.  In fact, even the most well-studied transfers, like the sigmoid, were not covered by the existing literature in this context.  Our aim was precisely to fill in this gap.
>
> Regarding technical novelty, specifically compared to Saha et al. (2021b): on a very high level, like Saha et al. we follow the approach of designing a gradient estimator using the dueling feedback and using it for running gradient descent.  However, our gradient estimation analysis is very different from that of Saha et al., due to the generality and nonlinearity of the transfer function in the feedback.  This can be seen in Lemma 3 and its proof, which analyzes the gradient estimator in the case of nonlinear transfer: roughly, we do this by inspecting the local polynomial behavior of the transfer around zero and working out the effect of the polynomial transformation on the expected value of the gradient estimator.  In the estimation process, the magnitude of the gradient gets distorted (see the $||\nabla f(w_t)||^{p-1}$ leading factor in the expected value) so we can no longer use the standard subgradient descent analysis, which we replace with a different approach akin to the analysis of normalized gradient descent.
>
> We agree a more detailed explanation of the technical novelty compared to Saha et al. should be included in the paper itself - we will improve this in the final version.
>
> > Other Comments Or Suggestions:
>
> Thank you for your suggestions, we will incorporate them in the final version of the paper.
>
> > “Since errors accumulate only along the path… will decrease by a $\log{⁡T}$ factor”
>
> We will have at the end of the run $T$ points, we would like to select the minimum of those $T$ corresponding function values. If we had exact comparisons, we could build a complete binary tree over the $T$ points, and compute the minimum by selecting at each node of the tree the minimum value between its descendents. With our actual noisy comparisons, we have “errors” of magnitude $\epsilon$ at each node of the tree, and if we sum these errors along the path to the minimal leaf, the error is amplified by a $\log(T)$ factor (which is the depth of the tree). Alternatively, we can start with a goal error of $\epsilon/\log(T)$ and end with a cumulative error of $\epsilon$.
>
> > “Can the assumption of $\rho(0)=0$ be relaxed?”
>
> It is worth recalling what the transfer function $\rho$ abstracts. One motivation is similar to Reinforcement learning from human feedback (RLHF), where the transfer function abstracts how human preference behaves given similarly-valued alternatives. The most natural one is to assume that when the alternatives are of similar values, humans select almost randomly. In our formal feedback model, this essentially implies that $\rho(0)=0$.
>
> Having said that, technically our algorithm works without modifications in the case where $\rho(0) \neq 0$, since it is agnostic to additive bias in $\rho$ (this can be seen from the statement of Lemma 9 in the appendix).
>
> > “Can the results be improved if $\rho$ is an odd function?”
>
> We are unaware of whether the bounds can be improved for odd transfer functions. In fact, our preliminary results were initially for odd functions and we were happy to see that they extend naturally to more general transfer functions, with the same rates, as we describe in the paper.

---

> > ### Comment · Reviewer_br4X · 2025-04-01
> >
> > I appreciate the authors for the response. The rebuttal revolves most of my concerns, especially the technical contributions. I have raised my score, and it would be great to see the discussion about the technical novelty compared with previous works in the revised version.

---

> > > ### Author Response · Authors · 2025-04-05
> > >
> > > We appreciate the reviewer's engagement and openness to revisit the initial concerns after reading our response. And thanks again for the thoughtful and constructive feedback---we will be sure to incorporate these discussions in the final version.

---

### Decision · Program_Chairs · 2025-05-01

**Decision:**

Accept (poster)

**Comment:**

Dear Authors,

Thank you for submitting your paper to ICML and for your contribution to the theoretical foundations of optimization. This work tackles the convex optimization setting under general preference-based comparisons and introduces a novel algorithm that achieves optimal convergence rates for both smooth and strongly convex functions.

The reviewers found the paper technically solid and appreciated the generalization of prior results. Reviewer br4X, after engaging with the rebuttal, increased the score, noting the clarity of the contributions. Other reviewers highlighted the elegance of the gradient estimator and the soundness of the theoretical analysis, while noting that the lack of experiments and applied motivation. Reviewers iszU and WhdB pointed out the importance of articulating practical examples and relaxing assumptions in theorems, which was addressed in the response.

Given the strength of the theoretical contributions, the soundness of the analysis, and the reviewers’ overall support, I am pleased to recommend acceptance to ICML 2025. I encourage you to clarify the assumptions and extend the discussion in the final version to help a broader audience understand the potential applications and interpretability of your results.

Congratulations on your contribution.

Best regards,
AC